

# Effects of thermal stress on amount, composition, and antibacterial properties of coral mucus

Rachel M. Wright[1,2], Marie E. Strader[2,3], Heather M. Genuise[2] and Mikhail Matz[2]

[1] Department of Genetics, Harvard Medical School, Boston, MA, United States of America
[2] Department of Integrative Biology, University of Texas at Austin, Austin, TX, United States of America
[3] Department of Ecology, Evolution, and Marine Biology, University of California, Santa Barbara, CA, United States of America

## ABSTRACT

The surface mucus layer of reef-building corals supports feeding, sediment clearing, and protection from pathogenic invaders. As much as half of the fixed carbon supplied by the corals' photosynthetic symbionts is incorporated into expelled mucus. It is therefore reasonable to expect that coral bleaching (disruption of the coral–algal symbiosis) would affect mucus production. Since coral mucus serves as an important nutrient source for the entire reef community, this could have substantial ecosystem-wide consequences. In this study, we examined the effects of heat stress-induced coral bleaching on the composition and antibacterial properties of coral mucus. In a controlled laboratory thermal challenge, stressed corals produced mucus with higher protein ($\beta = 2.1$, $p < 0.001$) and lipid content ($\beta = 15.7$, $p = 0.02$) and increased antibacterial activity (likelihood ratio = 100, $p < 0.001$) relative to clonal controls. These results are likely explained by the expelled symbionts in the mucus of bleached individuals. Our study suggests that coral bleaching could immediately impact the nutrient flux in the coral reef ecosystem via its effect on coral mucus.

## INTRODUCTION

Rising sea surface temperature has increased the global risk of coral bleaching, the breakdown of the symbiosis between a coral host and its algal symbiont, to alarming levels (*Hughes et al., 2018*). The direct impacts of bleaching on the animal host and algal symbiont are well studied. For example, coral bleaching has been shown to downregulate the expression of genes related to host immunity (*Pinzón et al., 2015*) and alter host metabolism (*Kenkel, Meyer & Matz, 2013*; *Rodrigues & Grottoli, 2007*). Symbionts (family Symbiodiniaceae) expelled during bleaching produce elevated amounts of reactive oxygen species but are otherwise physiologically similar to healthy endosymbionts (*Nielsen, Petrou & Gates, 2018*). Sustained coral bleaching often leads to host mortality and subsequent shifts in the reef community structure (*Harriott, 1985*). For example, coral bleaching and mortality causes shifts in reef-fish assemblage structure and alters recruitment success (*Booth & Beretta, 2002*; *Richardson et al., 2018*).

Corresponding author
Rachel M. Wright,
rachelwright8@gmail.com,
rachel_wright@hms.harvard.edu

Coral mucus is a complex mixture of proteins, lipids, and carbohydrates that is produced by mucocytes in the coral epidermal layer and secreted by coral surface tissues (reviewed in *Brown & Bythell, 2005*). Arabinose and glucose primarily compose the monosaccharide portion of coral mucus (*Wild, Woyt & Huettel, 2005*). Arabinose is likely a symbiont-specific contribution to coral mucus, as aposymbiotic, heterotrophic cold-water corals produce mucus nearly identical in composition to mucus from symbiotic warm-water corals, except for the presence of arabinose in the latter (*Wild et al., 2010*). Up to about half of the photosynthetically fixed carbon supplied by a coral's algal symbiont is expelled as mucus (*Crossland, 1987*; *Crossland, Barnes & Borowitzka, 1980*; *Davies, 1984*). This coral surface mucus layer acts as a defense against desiccation and pathogens (*Shnit-Orland & Kushmaro, 2009*), and is also released into the water column where it traps suspended particles and acts as an energy source for benthic communities (*Wild et al., 2004a*). Given the integral role of photosynthetically fixed carbon in mucus production of most reef-building corals, it is predicted that coral bleaching events will reduce mucus production and subsequently impact the flow of energy throughout the reef ecosystem (*Bythell & Wild, 2011*). However, one study has shown that acroporid corals exposed to an experimental heat stress released almost twice as much dissolved and particulate organic carbon relative to unchallenged controls (*Niggl et al., 2009*). One explanation for this finding is that increased mucus release in heat-stressed corals augments defense against pathogens when other immune defense mechanisms may be compromised by thermal challenge (*Palmer, Bythell & Willis, 2010*).

In the Florida Keys, annual mass bleaching events are predicted to begin by the mid-century (*Manzello, 2015*). Currently, multiple anthropogenic factors including thermal stress, increased storms, and disease outbreaks have led to a near 80% decline in reef cover in the Florida Keys since the 1980s (*Porter et al., 2001*; *Williams & Miller, 2011*; *Williams, Miller & Kramer, 2008*). In particular, corals in the genus *Acropora* have faced some of the most dramatic declines in this region (*Greenstein, Curran & Pandolfi, 1998*). The staghorn coral, *Acropora cervicornis*, has been selected as a focal species for multiple active restoration programs, such as the Coral Restoration Foundation, due to its relatively fast asexual growth through fragmentation (*Lirman et al., 2010*). As restoration efforts aim to replenish stands of *A. cervicornis*, it is critical to assess this species' greater role in coral reef ecosystem. This study aims to characterize coral mucus release, composition, and antibacterial activity during acute thermal stress in *A. cervicornis* as a reference to guide further research on the potential consequences of bleaching-mediated mucus changes on the larger coral reef ecosystem.

## MATERIALS & METHODS

### Corals

Fifteen *Acropora cervicornis* genets ($n = 3$ fragments per genet) were shipped from the Coral Restoration Foundation (Key Largo, Florida USA, Project ID CRF-2016-021) on 7 September 2016 to the University of Texas at Austin. Upon arrival, corals were immediately tagged with colored zip ties to uniquely identify each genet and allowed to recover for 12

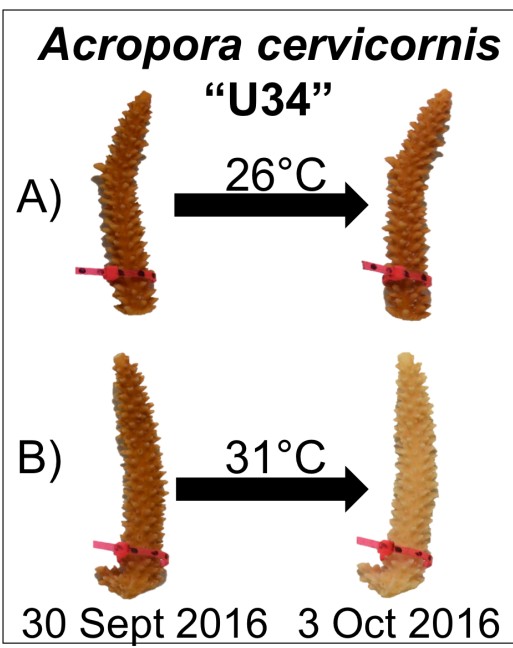

**Figure 1 Experimental design and representative coral image.** Corals were maintained in either (A) control (26 °C) or (B) experimental (31 °C) conditions for four days. Paling was observed for fragments in the experimental treatment, but not under control conditions.

days in artificial seawater (ASW; 30–31 ppt) at 25 °C under 12000K LED lights on a 12 h/12 h day/night cycle. Corals were fed weekly with Ultimate Coral Food (Coral Frenzy, LLC).

## Experimental conditions

Coral fragments were partitioned into experimental and control tanks. One genet (U10) experienced mortality during the recovery period, so only one U10 fragment remained when the experiment began. For all other genets, one fragment was placed in a control tank (26 °C) and one fragment was placed in an experimental tank. Any remaining fragments from the shipment of $n = 3$ per genet were retained in a holding tank, though many genets developed tissue loss or experienced damage on a single fragment during shipping. The single remaining U10 fragment was placed in the experimental tank. The temperature in the experimental tank was ramped from 26 °C to 31 °C over 33 h. High summer temperature in the Florida Keys often reaches 31 °C (*Manzello, 2015*). Therefore, a 31 °C heat treatment was chosen to represent an ecologically relevant stressor.

After corals had been exposed to experimental conditions for four days, corals appeared visibly pale relative to initial photographs and paired control fragments (Fig. 1). At this time, the temperature in the experimental tank was reduced to 26 °C over 6 h.

## Image analysis

Prior to the experiment, photographs of each fragment were taken using a Nikon D5100 camera. Images of the front and back of each fragment were taken using the same camera,

settings, and lighting each day of the experiment. Brightness values in images were measured for the front and back sides of each fragment using image analysis software (ImageJ, *Schneider, Rasband & Eliceiri, 2012*). Corals become brighter (paler) as their symbioses with pigmented algae break down. Therefore, changes in coral brightness reflect changes in algal densities (*Winters et al., 2009*). A standard curve of brightness values was constructed using standard Coral Health Charts that were included in each image. Briefly, brightness (white value intensity measured in unitless values ranging 0–255) was measured for a series of standard color cards (arbitrary values 1–6) in each photograph to construct a linear curve of brightness to compare fragments. Brightness values were standardized to color cards to normalize for any minor differences in lighting across days.

## Mucus collection

After the experiment, each coral fragment was placed within a pre-weighed 50 mL conical tube containing 5 mL ASW from the respective tank. Tubes were placed on their sides and secured to a gently rocking incubator plate (135 RPM, 28 °C) for 20 min, rotating the tubes every 5 min to ensure that all sides of the coral fragment were submerged in water. After rocking, fragments were inverted dry above the liquid in the conical for 20 min and lightly centrifuged (200 RPM) for 2 min to pull down mucus adhering to the surface of the coral, modified from (*Wild et al., 2004a*). The volume and mass of mucus from each fragment was measured and stored at −80 °C. Coral fragments were returned to their tanks. The mucus collection procedure was repeated six days later, exactly as described above. During mucus collection, algal cells were clearly visible in some samples. All mucus aliquots were briefly centrifuged at 3,500 $g$ for 2 min to sediment the algal pellet and decant algal cell-free mucus before the experiments described below.

## Mucus composition

Total protein was measured following the Coomassie (Bradford) Protein Assay Kit (Thermo Scientific, Waltham, MA, USA). Total carbohydrate was measured using the Total Carbohydrate Quantification Assay Kit (Abcam, Cambridge, UK). Total lipids were extracted and the dry mass of each mucus sample were measured. A standard curve was prepared using reagent grade cholesterol in a 2:1 chloroform:methanol mixture and an aliquot of 2:1 chloroform:methanol was added to each sample tube. After mixing, the solvent was evaporated from all standard and sample tubes on a heat block at 90 °C. Concentrated sulfuric acid was added to each tube, then incubated at 90 °C for 20 min. Samples were cooled, then plated in triplicate into wells of a 96-well plate. Background absorbance was measured at 540 nm. After incubating each sample with 50 µL of vanillin-phosphoric acid for 10 min, absorbance was measured again at 540 nm. The concentrations of protein, carbohydrate, and lipid in the mucus were estimated using standard curves constructed by measuring absorbance for known concentrations of protein, carbohydrate and lipids, per manufacturer's recommendations. All measurements were normalized to the volume of mucus expelled and the surface area of the fragment.

## Antibacterial activity

Cultures of laboratory *E. coli* (K-12) were grown overnight in Luria broth (LB), then washed twice in sterile ASW to remove remaining culture media. Coral mucus (140 µL) and washed *E. coli* culture (60 µL) was added to triplicate wells in 96-well plates. The covered plates were incubated at 37 °C for 12 h. Every 30 min the plate was shaken and the absorbance at 600 nm was measured. The plates were completely dry by the end of the incubation, so only the first 5 h, the latest time at which we observed liquid in the wells, are included in these analyses. Control wells included bacteria in LB or coral mucus without added bacteria to demonstrate bacterial growth or quantify endogenous mucus bacteria, respectively.

## Coral surface area

Fragment surface area was estimated using a 3D scanner and accompanying ScanStudioPro software (NextEngine, Santa Monica, CA, USA). Each scan was completed using a 360 degree scan with 16 divisions and 10,000 points/inch$^2$. Scans were then trimmed, polished to fill holes, fused and then surface area was estimated based on a size standard.

## Real-time quantitative PCR

The forward primer 5′-TCTGTACGCCAACACTGTGCTT-3′ and reverse primer 5′-AGTGATGCCAAGATGGAGCCT-3′ was used to amplify the *Acropora cervicornis* actin sequence as developed in (*Winter, 2017*). The forward primer 5′-GTGAATTGCAGAACTCCGTG-3′ and reverse primer 5′-CCTCCGCTTACTTATATGCTT-3′ was used to amplify the Symbiodiniaceae ITS2 sequence. Primer pair specificity was verified by gel electrophoresis and melt curve analysis of the amplification product obtained with *A. cervicornis* holobiont DNA. Primer efficiencies were determined by amplifying a series of four-fold dilutions of *A. cervicornis* holobiont DNA and analyzing the results using *PrimEff* function in the *MCMC.qpcr* package (*Matz, Wright & Scott, 2013*) in R. Briefly, $C_T$ (threshold cycle) results were plotted as CT vs. log$_2$[DNA], and amplification efficiencies (amplification factor per cycle) of each primer pair were derived from the slope of the regression using formula: efficiency $= 2^{-(1/\text{slope})}$ (*Pfaffl, 2001*).

Mucus aliquots were centrifuged to remove any cell debris. A 14 µL aliquot of coral mucus was combined with SYBR Green PCR Master Mix (Applied Biosystems), 1.5 µM forward and reverse primers, and water. The Roche LightCycler 480 system was used to carry out the PCR protocol (95 °C for 40 s, then 40 cycles of 60 °C for 1 min and 72 °C for 1 min) and detect the fluorescence signal.

## Statistics

All statistical analyses were performed in R (3.4.0, *R Core Team, 2016*). Bayesian models implemented using the MCMCglmm package (*Hadfield, 2010*) was used to explain variation in coral color and mucus composition, with treatment as a fixed effect and genotype as a random effect under a Gaussian distribution for coral color and metabolic assays and a Poisson distribution for marker gene counts. A nonlinear mixed-effects model followed by ANOVA was implemented using the nlme package (*Pinheiro et al., 2017*) was used for time-series analysis of antibacterial activity, with time, treatment, and their interaction

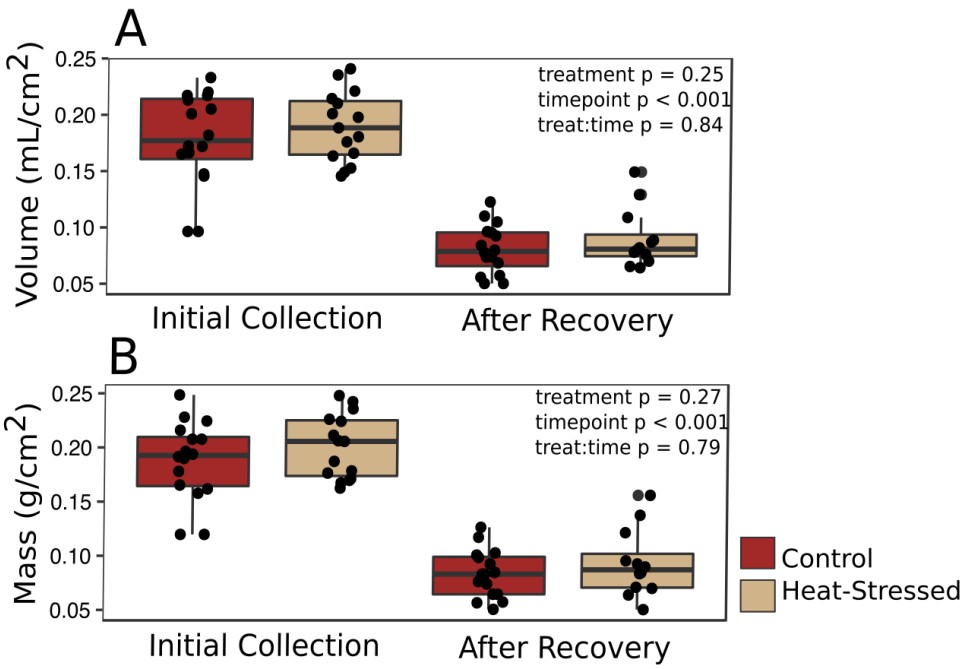

**Figure 2** **Mucus production.** Mucus was collected immediately after paling was observed ("Initial") and six days after the challenged corals were returned to control conditions ("Recovery"). The volume (A) and mass (B) of the recovered mucus was normalized to the surface area of the coral fragment. Red boxes represent fragments in control conditions; beige boxes represent heat-stressed fragments. Error bars represent standard error.

as fixed effects and plate well as a random effect. Experimental data and R scripts are included as Data S1 and S2, respectively. Data and scripts can also be accessed on GitHub: https://github.com/rachelwright8/bleached_coral_mucus.

## RESULTS

### Coral bleaching and mucus collection

Corals from both treatments produced similar amounts of mucus by volume ($\beta = 0.01$, $p = 0.25$, Fig. 2A) and mass ($\beta = 0.14$, $p = 0.27$, Fig. 2B). After a six-day recovery period, corals in both treatments produced significantly less mucus by volume ($\beta = -0.14$, $p < 0.001$, Fig. 2A) and mass ($\beta = -1.1$, $p < 0.001$, Fig. 2B) compared to the first time point. After four days in the experimental treatment at 31 °C, corals paled significantly compared to corals in the control condition ($\beta = -1.16$, $p < 0.001$, Fig. 3A).

### Mucus biochemistry

The mucus produced by heat-stressed fragments contained significantly more total protein ($\beta = 2.1$, $p < 0.001$, Fig. 3B) and total lipid ($\beta = 15.7$, $p = 0.02$, Fig. 3C). There was also a slight increase in carbohydrate content in mucus from heat-stressed corals, though this difference was not significant ($\beta = 0.64$, $p = 0.10$, Fig. 3D).

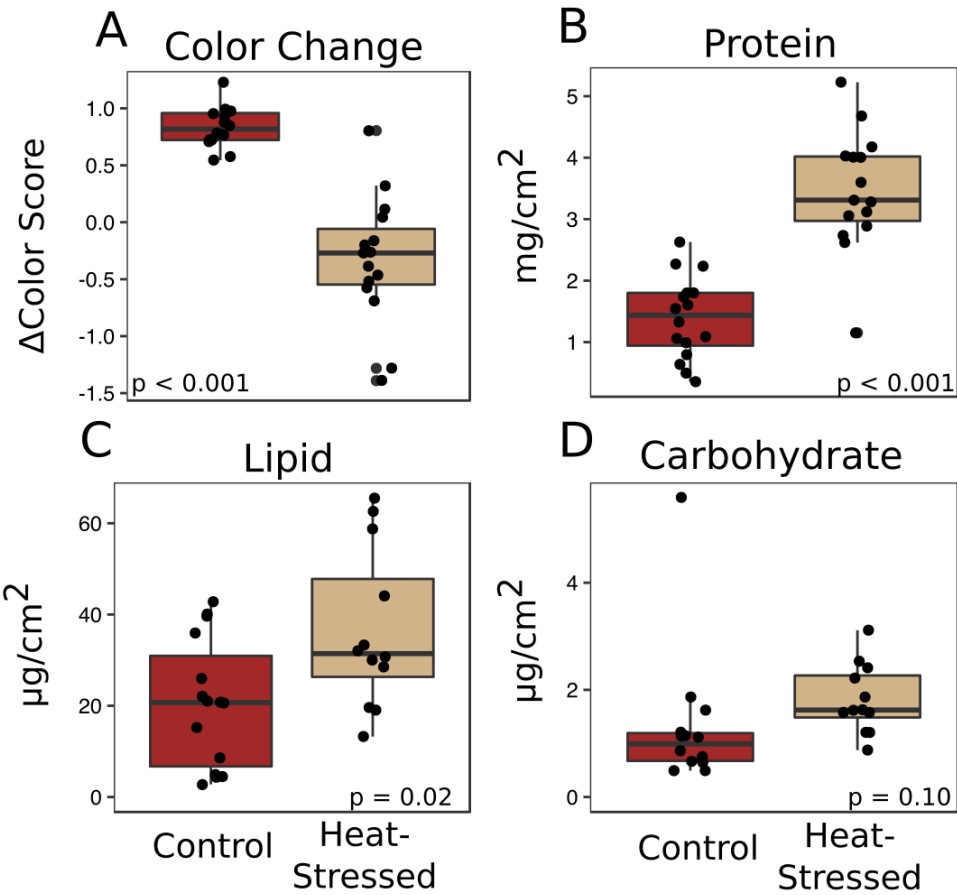

**Figure 3** **Effects of heat stress.** (A) Effect of treatment on coral color (decrease in color score indicates bleaching). (B–D) Effects on mucus composition: protein (B), in mg/cm² fragment surface area, and lipid (C) and carbohydrate (D), in μg/cm² fragment surface area. Red boxes represent fragments in control conditions; beige boxes represent heat-stressed fragments. Error bars represent standard error.

## Mucus antibacterial activity

Antibacterial activity increased in thermally stressed corals from the experimental treatment relative to healthy corals (likelihood ratio = 100, $p < 0.001$, Fig. 4). Optical density at 600 nm ($OD_{600}$), an absorbance measure that reflects bacterial density, decreased throughout the incubation period in all mucus samples. However, bacterial density declined significantly faster in mucus samples from heat-stressed corals. Bacterial density remained low throughout the sampling time period in negative control samples, which contained mucus from control corals with no added bacteria, suggesting that decreases in observed bacterial density were due to elimination of the experimentally added *E. coli* (Fig. S1). Positive control wells that contained *E. coli* in growth media with no coral mucus increased in OD, showing that the bacteria were alive and capable of growth under the experimental conditions (Fig. S1).

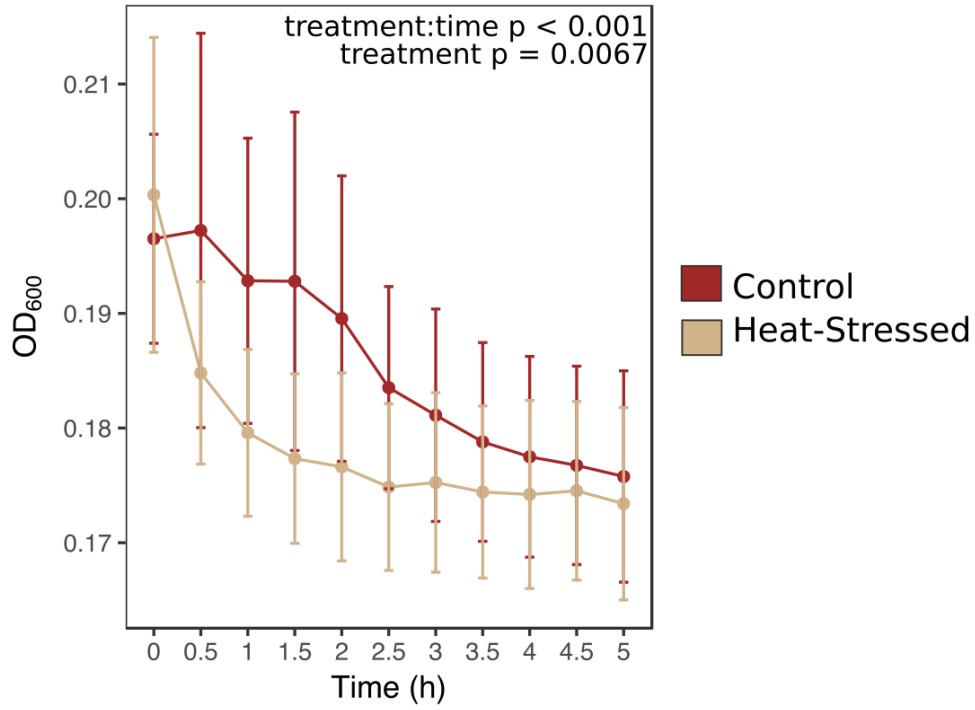

**Figure 4 Antibacterial activity of coral mucus.** Optical density (OD) at 600 nm indicates the density of inoculated bacteria in coral mucus samples. Red lines represent fragments in control conditions; beige lines represent heat-stressed fragments. Error bars represent standard error.

## Presence of host and symbiont DNA in mucus

Real-time quantitative PCR (qPCR) was performed to determine the relative abundances of coral- and symbiont-derived marker gene copies present in the mucus released by healthy and heat-stressed corals. Primers were designed to target a coral-specific actin gene and the Symbiodiniaceae ITS2 region. Presumably, copies of the coral-specific actin gene would represent lysed coral cells, while copies of the ITS2 region would represent material released from Symbiodiniaceae cells in the mucus. We did not detect any ITS2 sequences in the mucus of unchallenged corals (Fig. 5). Mucus released by heat-stressed coral fragments contained significantly more marker gene copies of both coral-derived (95% credible interval (CI) = 1.6–4.6, $p = 0.001$) and symbiont-derived sequences (95% CI [4.3–25.7], $p < 0.001$, Fig. 5).

## DISCUSSION

### Coral mucus stores take a long time to replenish

Mucus volumes released from both control and heat-stressed fragments during the initial collection ($0.18 \pm 0.04$ mL/cm$^2$, or about 1.8 L/m$^2$) are consistent with daily mucus release values previously reported for submerged acroporid corals (1.7 L/m$^2$ in *Wild et al., 2004a*). However, following a six-day recovery period, less than half of the original mucus volume was collected from both control and heat-stressed fragments ($0.18 \pm 0.04$ mL/cm$^2$ initial

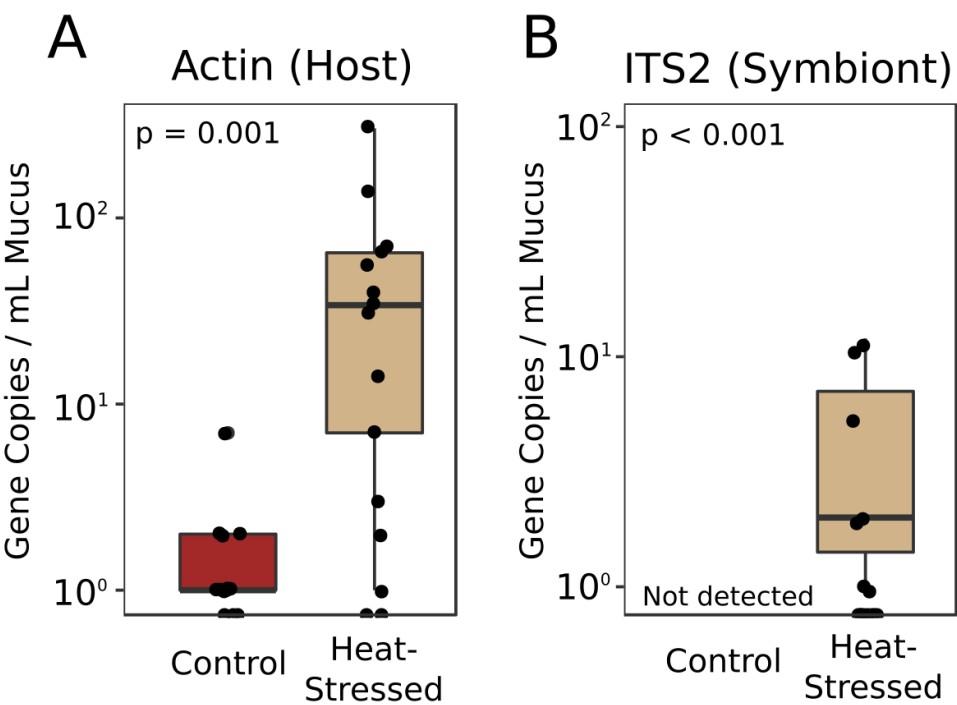

**Figure 5** **Symbiodiniaceae and coral DNA in mucus.** Real-time quantitative PCR detected *A. cervicornis* actin (A) or Symbiodiniaceae ITS2 (B) marker gene copies in the coral mucus of healthy and pale corals. Red boxes represent fragments in control conditions; beige boxes represent heat-stressed fragments. Error bars represent standard error.

collection *vs.* $0.08 \pm 0.02$ mL/cm$^2$ post-recovery, $p_{MCMCglmm} < 0.001$, Fig. 2A), suggesting that mucus stores were not completely replenished to the original volume in this amount of time. The methods used in this study to extract mucus left the coral nubbins completely dry. Therefore, the results presented here represent the total volume of stored mucus at the time of collection, which may not correspond directly to daily mucus release in natural settings. These results emphasize the importance of measuring mucus release over time to confidently estimate daily release rates and predict daily energetic flow throughout the reef ecosystem.

### Stressed corals produce mucus high in protein and lipid
We found no difference in the quantity of mucus produced by stressed corals compared to healthy corals after four days of heat stress or after a six-day recovery period (Fig. 2), suggesting that the quantity of mucus produced by a coral is relatively unaffected by thermal stress and that mucus stores cannot be replenished within a week. Given the energetic cost of producing mucus (*Riegl & Branch, 1995*), it is reasonable to predict that corals with low densities of autotrophic symbionts would produce less, or lower nutritional quality, mucus than healthy corals. These findings of enhanced protein and lipid content in mucus from heat-stressed corals relative to mucus produced by healthy corals (Fig. 3) support a previous study that found increased organic carbon release by thermally stressed corals (*Niggl et al., 2009*). Symbiodiniaceae store reserve energy as lipid

droplets and starch granules that are translocated from the algal membrane to coral cells in a healthy coral–algae symbiotic relationship (*Patton & Burris, 1983*). We observed a pellet of algal cells in the mucus of thermally stressed corals, but not in mucus produced by healthy corals. Though algal cells were pelleted and removed from all mucus collections, extracellular lipid droplets would remain in the mucus and could represent a potential explanation for the increased abundance of lipids in mucus from stressed corals. Likewise, proteins and lipids released from damaged Symbiodiniaceae and host cells would also be present in the mucus of stressed corals. This particular possibility is supported by finding both coral and Symbiodiniaceae DNA in the mucus of stressed corals (Fig. 5). Future studies should measure long-term effects of bleaching to determine the duration of this observed enrichment in coral mucus quality following thermal stress.

This short-term enrichment in nutrition could invoke shifts in trophic interactions and density dependent foraging behavior in coralivorous fish and invertebrates. In the Florida Keys, previous studies have observed increased predation intensity by corallivorous fishes and invertebrates as coral cover declines (*Baums, Miller & Szmant, 2003*; *Burkepile, 2012*). In addition, there is established prey preference asymmetries in the Caribbean, where *A. cervicornis* is already the preferred prey of a corallivorous gastropod (*Johnston & Miller, 2014*). Our results implicate that as heat-stressed corals release more nutrients into their mucus, this may drive stronger predation pressure, at least in bursts associated with the bleaching event itself. This may only maintain a benefit to higher trophic levels at very short time scales, as predation on coral reef dwelling fish is higher when they reside in bleached corals (*Coker, Pratchett & Munday, 2009*) and corallivorous fish are well known to suffer population declines in response to coral bleaching (*Pratchett et al., 2018*). Therefore, future studies should investigate how bleaching impacts predator–prey dynamics between corals and corallivorous fish and invertebrates.

## Stressed corals produce mucus with high antibacterial activity

This study found that mucus collected from stressed coral fragments eliminated bacteria faster than mucus from healthy fragments from matched genets (Fig. 4). Antibacterial activity of coral mucus is attributed to antimicrobial substances produced by commensal microbes living on the coral surface (*Nissimov, Rosenberg & Munn, 2009*; *Shnit-Orland & Kushmaro, 2009*). A previous study of mucus collected from *A. palmata* during a summer bleaching event in the Florida Keys found that mucus from bleached corals lacked antibiotic properties normally found in healthy coral mucus (*Ritchie, 2006*). The difference in findings between this study and the investigation of antibiotic activity in naturally bleached corals could be attributed to the timing of collections. Mucus in this study was collected as soon as corals became pale, whereas the previous study collected mucus after corals in the Florida Keys had been experiencing high levels of thermal stress and bleaching for about a month according to NOAA reports of the thermal stress event (*Eakin et al., 2010*). Long-term thermal stress is known to promote coral disease by altering bacterial pathogenicity and host susceptibility (*Bruno et al., 2007*; *Maynard et al., 2015*). In our short-term bleaching conditions, the increased protein and lipid content in the mucus (Figs. 3B–3C) may have temporarily improved the antibacterial activity of commensal

microbes that exist in the coral mucus (*Krediet et al., 2013*). Though the mechanism is unclear, Symbiodiniaceae do appear to play a role in a coral's ability to manage immune stress and regulate microbial communities (*Littman, Bourne & Willis, 2010*; *Wright et al., 2017*).

Other studies that have investigated bacterial growth in coral mucus have found that mucus supports the growth of bacterial species found on corals (*Wild et al., 2004b*), such as *Pseudoalteromonas* and *Vibrio* spp. (*Allers et al., 2008*). Both *Pseudoalteromonas* and *Vibrio* are commonly observed in corals, though they are usually associated with coral stress and disease suggesting that these bacteria employ mechanisms to resist antimicrobial properties of coral mucus (*Ritchie, 2006*) that our lab *E. coli* strain, which was naïve to coral mucus, presumably lacks. This difference in bacterial species could explain the discrepancy in findings. Additionally, the previous experiments measured bacterial growth over 40–50 h. It is possible that we would have found similar results had we measured bacterial growth beyond 5 h.

## CONCLUSIONS

Our results show that thermal stress does not significantly affect the volume of mucus produced by *A. cervicornis* immediately following a bleaching event. Stressed corals produced mucus with higher protein content, higher lipid content, and increased antibacterial activity relative to unstressed controls. Additional lipids and proteins likely come from Symbiodiniaceae and host cells damaged during bleaching rather than from additional investment by the coral host. Short-term nutritional enrichments of mucus released from bleaching corals could promote growth of heterotrophic microbes at the lowest trophic levels of marine ecosystems and thus cause large-scale shifts in a reef's nutrient cycle. Changes in the nutritional composition and antibacterial properties of the mucus should influence coral-associated microbes that contribute to coral disease susceptibility. Future experiments should investigate longer-term effects of thermal stress on mucus production and content to further investigate reef-wide consequences of coral bleaching.

## ACKNOWLEDGEMENTS

We thank the Coral Reef Foundation for providing coral specimen and Sarah Davies for measuring coral surface areas.

### Funding

This work was funded by a 2016 PADI grant #21956 awarded to Marie Strader and a University of Texas Co-op Undergraduate Research Fellowship awarded to Heather Genuise. The funders had no role in study design, data collection and analysis, decision to publish, or preparation of the manuscript.

## Grant Disclosures

The following grant information was disclosed by the authors:
2016 PADI: #21956.
University of Texas Co-op Undergraduate Research.

## Competing Interests

The authors declare there are no competing interests.

## Author Contributions

- Rachel M. Wright conceived and designed the experiments, performed the experiments, analyzed the data, contributed reagents/materials/analysis tools, prepared figures and/or tables, authored or reviewed drafts of the paper, approved the final draft.
- Marie E. Strader conceived and designed the experiments, performed the experiments, analyzed the data, contributed reagents/materials/analysis tools, authored or reviewed drafts of the paper, approved the final draft.
- Heather M. Genuise performed the experiments, analyzed the data, contributed reagents/materials/analysis tools, authored or reviewed drafts of the paper, approved the final draft.
- Mikhail Matz conceived and designed the experiments, contributed reagents/materials/analysis tools, authored or reviewed drafts of the paper, approved the final draft.

## Data Availability

The data and R scripts are available at https://github.com/rachelwright8/bleached_coral_mucus.

## Supplemental Information

Supplemental information for this article can be found online at http://dx.doi.org/10.7717/peerj.6849#supplemental-information.

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
