# Peer review of "Effects of thermal stress on amount, composition, and antibacterial properties of coral mucus"

_PeerJ, doi:10.7717/peerj.6849_

## Round 0.1 · original submission · Major Revisions

Your manuscript has now been reviewed by two experts in the field of coral reef microbiology, both of whom provided excellent and detailed review of your manuscript. Both provided several recommendations for improvement, all of which I concur with as potentially improving the manuscript; I request that with your revised manuscript you submit a point-by-point response detailing how you have incorporated or otherwise considered the reviewers' input. In drafting your response to reviewers please consider the reviewers' time and detail the edits you make with both line numbers and in-line quotes to facilitate their review of the revision.

Reviewer 1 ·

Basic reporting

The manuscript by Wright et al. addresses the topic of coral mucus production and composition under heat stress, a current threat to coral reefs worldwide. The authors find that mucus of stressed corals has higher lipids and protein content as well as more Symbiodinium and coral cell-free DNA. The authors also find that mucus released by stress corals has unexpected higher anti-microbial activity against E. coli K-12 compared to the mucus released by healthy corals. The manuscript contains interesting and valuable for the field of coral microbiology, but few minor clarifications and comments are required to verify the validity of the findings and improve the presentation of the results.

Experimental design

For instance, the estimate of anti-microbial activity was done by inoculating mucus samples with E. coli cells that were previously grown, but how do you know that bacterial grow measurements correspond to E. coli and not mucus-associated bacteria? Did you filter mucus samples?

Validity of the findings

no comment

Additional comments

In line with your results, I would argue that the increased anti-microbial activity could potentially be explained as a mechanism to further prevent bacterial colonization triggered by increased protein and lipid contents?

Figure 4. Could you add absorbance values on the y-axis? Why did you only show 5 if the experiment was done over 12 hours?

Figure 5. Do “counts” refer to gene copy number?

L187: Once you set your tolerance level, the result is either significant or it isn’t. Avoid “marginally significant increase”, if p-value>0.05

L198: Maybe “marker gene copies” instead of “DNA sequences”?
More lipids, more proteins more antimicrobial activity

L203: Can you indicate how many gene copies you found in both conditions? “Few” gene copies is not correct.

L209: please indicate what the numbers refer to, e.g. means of untreated corals.

L212: Again, what does the 46.8±13.2 refer to? If that refers to the percentage of mucus produced versus the original mucus volume, don’t you think it will be more informative to add the mean volume produced?

·

Basic reporting

- Language:
o Clear and unambiguous language is used. Current phrasing could be improved in lines 214-216, 246-251 (clarify references), 267-270 (simplify). Further suggestions on language are:
o Lines 31, 87: meteorological parameters (here: temperature) are used in singular (e.g. “Summer/water temperature…reaches…”)
o Lines 76 (°C), 76 (h), 82 (°C), …, 87 (hours), 92 (°C, hours), …, 209 (±), 212 (%) and some more. I recommend to increase consistency in the use of units (h, hours) and spaces in front of units/symbols (currently: °C no space, mL space, = space, ± no space).
o 119/120: dry mass instead of dry weight
o Lines 48/49: double ((
o Line 132: introduce LB
o Line 236: of
o Line 148: Winter (2017)

- Intro and background:
o Important literature such as “Niggl, W., Glas, M., Laforsch, C., Mayr, C., & Wild, C. (2009) First evidence of coral bleaching stimulating organic matter release by reef corals.” is missing so far and should thus be considered. Also, reference “Wild C, Woyt H, Huettel M (2005) Influence of coral mucus on nutrient fluxes in carbonate sediments. Marine Ecology Progress Series 287: 87-98” and “Wild C, Naumann MS, Niggl W, Haas AF (2010) Carbohydrate composition of mucus released by scleractinian warm and cold water reef corals. Aquatic Biology 10: 41-45” should be considered in introduction and discussion, because in these publications compositional analyses of coral mucus are provided. For the discussion about antibacterial properties of corals mucus it would be beneficial to compare findings of this study with findings of “Allers E, Niesner C, Wild C, Pernthaler J (2008) Microbes enriched in seawater after the addition of coral mucus. Applied and Environmental Microbiology 74(10): 3274-3278” and “Wild C, Rasheed MY, Werner U, Franke U, Johnstone R, Huettel M (2004) Degradation and mineralization of coral mucus in reef environments. Marine Ecology Progress Series 267: 159-171”.
o Potentially also helpful: “Lee, S. T., Davy, S. K., Tang, S. L., Fan, T. Y., & Kench, P. S. (2015). Successive shifts in the microbial community of the surface mucus layer and tissues of the coral Acropora muricata under thermal stress.”
o Additional references would be useful in lines 44/45, 60/61, 62 in order to provide more profound background.
o Lines 54/55: As they are a major component of the further analysis and discussion, a more profound introduction of the nutritional composition and antibacterial properties of mucus would be helpful.
o Lines 23, 33: When referring to ecosystem functions, it would be great to go into more detail what this means – a) L. 23: what does bleaching mean for the system? and b) L. 33: which ecosystem functions are affected by (expelled) nutrient composition? Therefore, shift lines 32/33 (“highlighting…..they support”) towards the end of this paragraph. There, it is possible to provide more detail on the (above mentioned) affected ecosystem functions. Authors state a need to understand impacts of bleaching on coral communities (L. 33), but name “numerous studies” showing them (mortality on community structure (L. 39/40)).
o Lines 17-21: The beginning of the abstract could be shortened. More emphasis should be put on the justification of study and the gaps of knowledge.
o Lines 23-25: This section of the abstract should be enlarged and underlined by some numbers that illustrate key results.

- Structure and standards:
Overall structure conforms to PeerJ standards.
“&” could be used in more than one authored references as suggested in PeerJ Standards.
- Figures:
Consistency in color use and style is generally good, but legends (branches with “healthy” and “pale”) are questionable because
- general terms healthy and pale (see general comments)
- term varies regarding the statistical compartment. There, the heat stress effect was called treatment. In the legend it is called “healthy” and “pale”. But in the end, it is thermal stress. So maybe refer to it as treated/untreated or heat stressed/control. Also see Fig. 3.
- Coral shape not needed for legend
o Figure 2:
Better not use “weight”. Is it dry mass?
Please mention the colors and their meaning in the description
o Figure 3:
The A section of the graph should be separated from the remaining compartments. The measure of heat stress is the fundamental distinction/treatment in this study (“healthy” and “pale”). Accordingly, it is not a comparable response parameter on the same level as the others. It should be presented in the beginning of the results section, potentially related with Fig. 1. So far, it appears inconsistent to use color change as a result of thermal stress, while differentiating all graphs according to coloration.
o Figure 4:
What do the bars represent? Please specify in the description.
Time in h
Specify the scale. Does not the relative absorbance have a unit?
Raw data: is provided with files and Rscript. Thanks for having that all in good structure.

Experimental design

- The scope of PeerJ seems to be met.
- The research questions should be more sharply defined. Lines 65-67: More detail on the measures used and the relation to previously mentioned gaps of knowledge, as well as the potentially outlined ecosystem functions/implications would be beneficial. This can be a brief repetition of previously mentioned gaps in knowledge, but helps to justify study. Also, it would smoothen the path coming from the Acropora topic.
- The investigation appears of high technical and ethical standard.
- Methods are overall described with sufficient detail. More information should be added in lines 100/101, 111/112, 114/115, 128 to better understand the procedure and allow for replication.
Lines 165-168 should include a general sentence on model assumptions and what statistical approach was used (repeated measures anova?)
Lines 153/154 belongs to statistical analysis
Background information on nutrient conditions of the incubations/ASW would be very useful here.

Validity of the findings

Even though not including all relevant literature (e.g. Niggl et al. 2009), this work provides very interesting results.
Line 260: Towards the end of the discussion a short outlook referring to the ecosystem implications would be great. The reader may be interested to know what results could mean for the ecosystem, as ecosystem function is mentioned in abstract, e.g. short-term increase / release of more protein and lipid components – who could benefit from that?
Line 268: Please mention that this influence on the reefs nutrient cycle is at least short-term – because discussion highlights that it could be linked to the zooxanthellae expelled.
Further references should be given in 213-218, 255/256 to relate findings to current knowledge and underline what findings could mean.
Line 221: Each section of the discussion should begin with a short reminder what the key related study results according to the specific topic of interest / research question were. Accordingly, lines 221-223 should be placed at the end of line 226.
Line 242: “Surprisingly” suggests that there was an expectation beforehand – but there was no clearly stated hypothesis about that. This also refers to being a bit more detailed in the research question.
Line 185: “although” sounds like contradicting to an expectation, which should be discussed later and not in the results.

Additional comments

This is a solid and interesting study that deserves publication. However, the manuscript would highly benefit from a thorough overview of relevant literature.
Also, it appears inconsistent to use a response parameter (color) to differentiate between treatments. There are not many physiological measurements that confirm that control corals were “healthy”. They were not under thermal stress. If using that term, it would be better to define them as healthy according to Winters et al. (2009). The pale “treatment” individuals can be stated like this, but only when separated from the remaining response parameters and shown in the beginning of the results section. As the title of the study is “the effects of thermal stress”, it is certainly more appropriate to remain the color as a response parameter (also in Figure 3), but refer to the treatment as heat stress/thermally stressed and controls/unstressed.
Furthermore, some more detail in abstract, a more clear formulation of research questions, and provision of background on ecosystem implications would make the study more useful for the reader.

---

## Round 0.2 · accepted · Accept

Both reviewers were clearly appreciative of the work put into the revisions and we are pleased to accept this work.

# Reviewer 1 ·

Basic reporting

no additional comments

Experimental design

The authors have clarified methodological concerns raised on the first version of the manuscript as well as improved the narrative and presentation of the results.

Validity of the findings

no additional comments

Additional comments

No additional comments

·

Basic reporting

Much better now.

Experimental design

No comment.

Validity of the findings

Much better now.

Additional comments

The authors have addressed all my previous comments sufficiently so that I can recommend acceptance of this manuscript.